# Dexterity from Touch: Self-Supervised Pre-Training of Tactile Representations with Robotic Play

**Irmak Guzey**[1,†]    **Ben Evans**[1]    **Soumith Chintala**[2]    **Lerrel Pinto**[1]

New York University[1], Meta[2]

tactile-dexterity.github.io[*]

**Abstract:** Teaching dexterity to multi-fingered robots has been a longstanding challenge in robotics. Most prominent work in this area focuses on learning controllers or policies that either operate on visual observations or state estimates derived from vision. However, such methods perform poorly on fine-grained manipulation tasks that require reasoning about contact forces or about objects occluded by the hand itself. In this work, we present T-DEX, a new approach for tactile-based dexterity, that operates in two phases. In the first phase, we collect 2.5 hours of play data, which is used to train self-supervised tactile encoders. This is necessary to bring high-dimensional tactile readings to a lower-dimensional embedding. In the second phase, given a handful of demonstrations for a dexterous task, we learn non-parametric policies that combine the tactile observations with visual ones. Across five challenging dexterous tasks, we show that our tactile-based dexterity models outperform purely vision and torque-based models by an average of 1.7X. Finally, we provide a detailed analysis on factors critical to T-DEX including the importance of play data, architectures, and representation learning.

**Keywords:** Tactile, Dexterity, Manipulation

## 1 Introduction

Humans are able to solve novel and complex manipulation tasks with small amounts of real-world experience. Much of this ability can be attributed to our hands, which allow for redundant contacts and multi-finger manipulation. Endowing multi-fingered robotic hands such dexterous capabilities has been a long-standing problem, with approaches ranging from physics-based control [1] to simulation to real (sim2real) learning [2, 3]. More recently, the prevalence of improved hand-pose estimators has enabled imitation learning approaches to teach dexterity, which in turn improves sample efficiency and reduces the need for precise object and scene modelling [4, 5, 6].

Even with improved algorithms, teaching dexterous skills is still quite inefficient, requiring large amounts of demonstration data and training [5, 2]. While algorithmic improvements in control will inevitably lead to improvements in dexterity over time, an often overlooked source of improvement lies in the sensing modality. Current dexterous robots either use high-dimensional visual data or compact states estimated from them. Both suffer significantly either when the task requires reasoning about contact forces, or when the fingers occlude the object being manipulated. In contrast to vision, tactile sensing provides rich contact information while being unaffected by occlusions.

So, why is tactile-based dexterity hard to achieve? There are three significant challenges. First, tactile sensors are difficult to simulate, which limits the applicability of sim2real based methods [2, 7] to binary contact sensors [8]. Second, for many commonly available tactile sensors, precise calibration of analog readings to physical forces is difficult to achieve [9]. This limits the applicability of

---

[*†]Correspondence to irmakguzey@nyu.edu.

7th Conference on Robot Learning (CoRL 2023), Atlanta, USA.

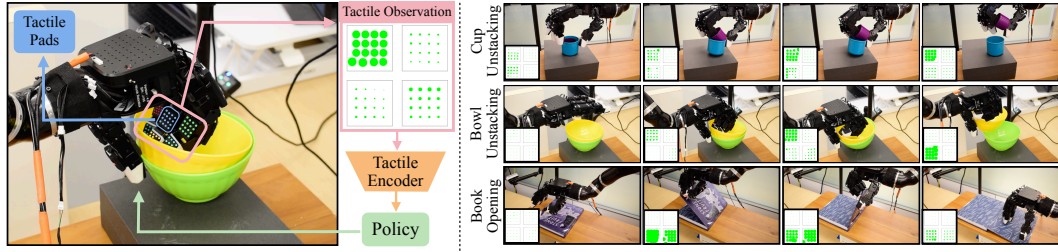

(a) Framework for Tactile-based Dexterity    (b) Visualization of T-Dex rollouts on three selected robot tasks

Figure 1: T-DEX learns dexterous policies from high-dimensional tactile sensors on a multi-fingered robot hand. Combined with vision, our tactile representations are crucial to learn fine-grained manipulation tasks.

physics-based control. Third, for multi-fingered hands, tactile sensors need to cover a larger area compared to two-fingered hands. This increases the dimensionality of the tactile observation, which in turn makes learning-based approaches inefficient. A common approach to alleviate this challenge in vision-based learning is to use pretrained models that encode high-dimensional images to low-dimensional representation. However, such pretrained models do not exist for tactile data.

In this work, we present T-DEX, a new approach to teach tactile-based dexterous skills on multi-fingered robot hands. To overcome issues in simulating tactile sensors and calibration, we use an imitation framework that trains directly using raw tactile data obtained by teleoperating the robot. However, directly reasoning about actions from raw tactile data would still require collecting large amounts of demonstrations. To address this, we take inspiration from recent works in robot play [10], and pretrain our own tactile representations. This is done by collecting 2.5 hours of aimless data of object manipulation. Tactile data collected through this play is used to train tactile encoders through standard self-supervised techniques, mitigating the need for exact force calibration.

Given this pretrained tactile encoder, we use it to solve tactile-rich dexterous tasks with just a handful of demonstrations: 6 demonstrations per task, corresponding to under 10 minutes of demonstration time. To achieve imitation with so few demonstrations, we employ a non-parameteric policy that retrieves nearest-neighbor actions from the demonstration set. Such a policy provides significant improvements over fully parameteric policies (Table 1). Importantly, this enables combining tactile encodings with other sensor modalities such as vision without additional training or sensor fusion. This ability to combine touch with vision makes T-DEX compatible with tasks that require visual sensing for coarse-grained manipulation and tactile sensing for fine-grained manipulation.

We evaluate T-DEX across five challenging tasks such as opening a book, bottle cap opening, and precisely unstacking cups (see Figure 1). Through a large-scale experimental study of over 50 hrs of robot evaluation we present the following insights:

1. T-DEX improves upon vision-only and torque-only imitation models with over a 170% improvement in average success rate (Section 4.2).

2. Play data significantly improves tactile-based imitation, with an average of 58% improvement over tactile models that do not use play data (Section 4.3).

3. Ablations on different tactile representations and architectures show that the design decisions in T-DEX are important for high performance (Section 4.4).

Robot videos and qualitative studies of T-DEX are best viewed on `tactile-dexterity.github.io`.

## 2   Related Work

Our work builds on several prior ideas in tactile sensing, representation learning and imitation learning. For brevity, we describe the most relevant below and present more discussion in Appendix A.

**Tactile Sensing in Dexterity:** Multi-fingered robot control has been studied extensively [11, 12, 13]. Initial work focuses on physics-based modelling of grasping [14, 15] that often used contact force estimates to compute grasp stability. However, contact estimates derived from motor torque only give point estimates and are susceptible to noise due to coupling with the hand's controller. To give robots a sense of touch, many tactile sensors have been created for enhancing robotic sensing [16, 17, 18]. Prominently, the GelSight sensor has been used for object identification [19], geometry sensing [20], and pose estimation [21]. However, since GelSight requires a large form factor, it is difficult to cover an entire multifingered hand with it. Instead, 'skin'-like sensors [22] and tactile pads can cover entire hands, yielding high-dimensional tactile observations for dexterity. In this work, we use the XELA uSkin [23] sensors to cover our Allegro hand.

Learning-based approaches have been employed to leverage high-dimensional readings from tactile sensors for a variety of applications such as grasping and manipulation with two-fingered grippers [24, 25, 26], object classification [27] and 3d shape detection [28]. However, these prior works differ from T-DEX in two key ways. First, such tactile learning methods have not been applied to multifingered hands. Second, the tactile representations learned in these works require large amounts of task-centric data for each task. On the other hand, T-DEX uses a large amount of task-agnostic play data, which enables learning tasks with small amounts of data per task. Concurrent to our work, binary tactile sensing has shown success for in-hand manipulation [8] through sim2real training. However, its application to high-dimensional tactile data is under explored.

**Representations in Offline Imitation** Imitation Learning (IL) allows for efficient training of skills from data demonstrated by an expert. Given a set of demonstrations, offline imitation methods such as Behavior Cloning (BC) use supervised learning to learn a policy that outputs actions similar to the expert data [29, 30, 31, 32, 33]. However, such methods often require demonstrations on the order of hundreds to thousands trajectories and collecting the same quantity of data for dexterous tasks is difficult due to cognitive and physical demands of teleoperating multi-fingered hands. To learn with fewer demonstrations in high-dimensional action spaces, non-parametric approaches such as nearest neighbors have shown to be more effective than parametric ones [4, 5]. Although efficient, non-parametric learning approaches require good representations of the environment. Several prior work have looked at learning visual features using self-supervision [34, 35, 5, 36]. We build on this idea to tactile observations and train tactile features using our human-generated robot play data.

**Exploratory and Play Data:** Since task-specific data can be expensive to collect, a number of works have examined leveraging off-policy data to improve task performance. Previous work has used play data to learn latent plan representations [37] and to learn a goal-conditioned policy [38]. Recent work in offline RL has noted that including exploratory data improves downstream performance [39] and that actively straying away from the task improves robustness [40]. These findings are paralleled by studies on motor development in humans. 3-5-month old infants spontaneously explore novel objects [41] and 15-month-old infants produce the same quantity of locomotion in a room with or without toys [42]. Given these motivating factors, we opt to leverage a play dataset of imperfect, tactile-rich interactions in order to improve our representations and task performance.

## 3 Tactile-Based Dexterity (T-DEX)

T-DEX operates in two phases: pretraining from task-agnostic play data and downstream learning from a few task-specific demonstrations. In the pretraining phase, we begin by collecting a diverse, contact-rich play dataset from a variety of objects by teleoperating the robot (see Appendix B.1 for the setup). Once collected, we use self-supervised learning (SSL) [43] algorithms on the play data to learn an encoder for tactile observations. In the downstream learning phase, a teleoperator collects demonstrations of solving a desired task. Given these demonstrations, non-parametric imitation learning is combined with the pretrained tactile encoder to efficiently learn dexterous policies. See Figure 3 for a high-level overview of our framework. Details of individual phases follow.

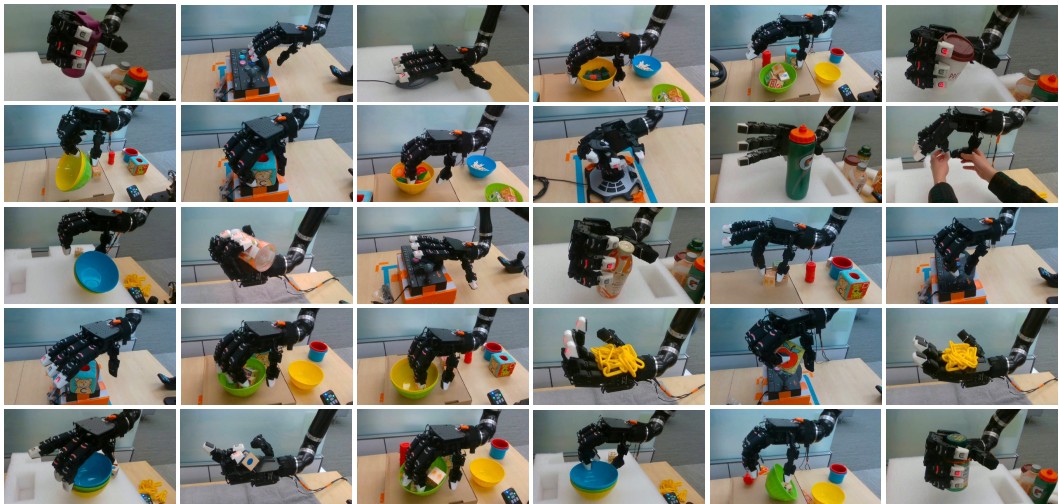

Figure 2: Visualization of some of the play tasks that uses grasping, pinching and other manipulation tasks.

## 3.1 Phase I: Pre-Training Tactile Representations from Play

**Play Data Collection:** The play data is collected from a variety of contact-rich tasks including picking up objects, grasping a steering wheel, and in-hand manipulation which are illustrated in Figure 2. We collect a total of 2.5 hours of play data, including failed examples and random behavior. Because the image and tactile sensors operate at 30Hz and 100Hz, respectively, we sub-sample the data to about 10Hz to reduce the size of the dataset. We only include observations whenever the total changed distance of the fingertips and robot end effector exceed 1cm, reducing the dataset from 450k frames to 42k. This filters out the similar states when the robot is still, which could potentially bias the SSL phase. All this data, along with camera streams, is publicly released on our project website.

**Feature Learning:** To extract useful representations from the play data we use Bootstrap your own Latent (BYOL) [44], an SSL method which tries to learn informative representations from raw observations. We treat the tactile sensor data as an image with one channel for each axis of force. Each of the finger's 3-axis 4x4 sensors are stacked into a column to produce a 16x4 image for the fingers and a 12x4 image for the thumb. These images are then concatenated to produce a three-channel 16x16 image with constant padding for the shorter thumb. A visualization of the tactile images can be seen in Figure 3. We pre-train an Alexnet [45] encoder on these tactile images from the play data using BYOL and use that encoder to extract tactile features in downstream tasks. More information on BYOL and our use of it on tactile readings can be found in Appendix C.1 and D.1.

## 3.2 Phase II: Non-parametric Learning

**Demonstration Collection:** Six demonstrations are collected for each task by a teleoperator using the Holodex [5] framework. Because the nature of our tasks are contact-dependent and the human operator does not receive tactile feedback, there is a relatively high failure rate while collecting demonstrations. Although successful demonstrations correspond to at most 10 minutes of robot time, it requires up to 60 minutes of collection time in order to get successful demonstrations which results in 5/6 failure rate in teleoperation. Success of a task is determined by visual inspection of robot scene by the teleoperator. This highlights the importance of using tactile feedback for dexterous manipulation along with the necessity of learning from few task-specific demonstrations.

**Visual Feature Learning:** Many dexterous tasks require coarse-grained information about the location of the object to be manipulated. This necessitates incorporating vision feedback as tactile observations are not meaningful when the hand is not touching the object. To do this, we extract visual features using standard BYOL augmentations on the images collected from demonstration

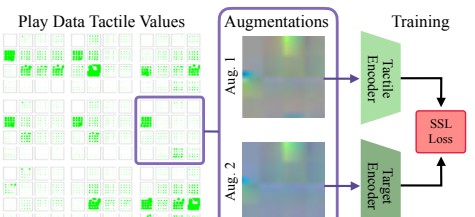
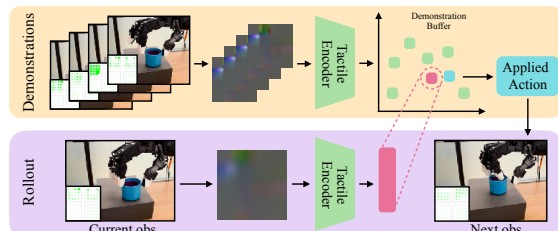

Figure 3: An overview of the T-DEX framework. Left: we train tactile representations using BYOL on a large play dataset. Right: we leverage the learned representations using nearest neighbors imitation.

data. The views for each task are significantly different, so we did not observe a benefit from including the play data in the visual representation learning. Similar to prior work [4, 5], we start with a ResNet-18 [46] architecture that has been pre-trained on ImageNet [47] classification.

**Downstream Imitation Learning:** Our action space consists of both the hand pose, specified by 16 absolute joint angles, and the robot end effector position and orientation, specified by a 3-dimensional position and a 4-dimensional quaternion. Due to both the high-dimensional action and observation spaces, parametric methods struggle to learn quality policies in the low-data regime. To mitigate this, we use a nearest neighbors-based imitation learning policy [35] to leverage our demonstrations. For each tuple of observations and actions in the demonstrations $(o_i^V, o_i^T, a_i)$, we compute visual and tactile features $(y_i^V, y_i^T)$ and store them along-side the corresponding actions. Since the scales of the two features are at different, we scale both features such that the maximum distance in the dataset for each feature is 1. At test time $t$ given $o_t$, we compute $(y_t^V, y_t^T)$, find the datum with the lowest total distance, and execute the action associated with it.

## 4 Experiments

We evaluate T-DEX on a range of tasks that are designed to answer the following questions: (a) Does tactile information improve policy performance? (b) How important is play data to our representations? (c) What are important design choices when learning dexterity from tactile information?

### 4.1 Experimental Overview

**Description of Dexterous Tasks:** We examine five dexterous contact-rich tasks that require precise multi-finger control. We show the robot rollouts for the selected 3 of the tasks in Figure 1. We describe them in detail in Appendix B.2, showcase the contribution of each taxel to tasks in Appendix C.2 and visualize the rollouts for them in Figure 9. To evaluate various models for dexterity, we first collect six demonstrations for each task in which the object's configuration is varying inside a 10x15cm box. Models are then evaluated on new configurations in the convex hull of demonstrated ones. This follows the standard practice of evaluating representations for robotics [48, 49, 50]. Additional experimental details exist in Appendix B.

**Baselines for Dexterity:** We study the impact of tactile information on policies learned through imitation, comparing against a number of baselines. Here, we briefly describe these baselines. Additional baseline and model details can be found in Appendix B.3-D.

1. *BC [51]*: We train a neural network end-to-end to map from visual and tactile features to actions.

2. *NN-Image [35]*: We perform nearest neighbors with the image features only.

3. *NN-Torque [52]*: We perform nearest neighbors with the output torques from our PD controller and visual representations.

4. *NN-Tactile*: Nearest neighbors with only the tactile features trained on play data.

Table 1: Real-world success rate of the learned policies

|  | BC | NN-Im | NN-Tac | NN-Task | NN-Tor | BET | IBC | GMM | **T-DEX** |
|---|---|---|---|---|---|---|---|---|---|
| Joystick | 0/10 | 4/10 | 6/10 | **8/10** | 7/10 | 0/10 | 0/10 | 0/10 | **8/10** |
| Bottle | 0/10 | 0/10 | **6/10** | 3/10 | 3/10 | 0/10 | 0/10 | 0/10 | **6/10** |
| Cup | 0/10 | 0/10 | 0/10 | 4/10 | 2/10 | 0/10 | 0/10 | 0/10 | **8/10** |
| Bowl | 0/10 | 2/10 | 2/10 | 3/10 | 4/10 | 0/10 | 0/10 | 0/10 | **7/10** |
| Book | 0/10 | 5/10 | 0/10 | 6/10 | 3/10 | 0/10 | 0/10 | 0/10 | **9/10** |
| Average | 0/10 | 2.2/10 | 2.8/10 | 4.8/10 | 3.8/10 | 0/10 | 0/10 | 0/10 | **7.6/10** |

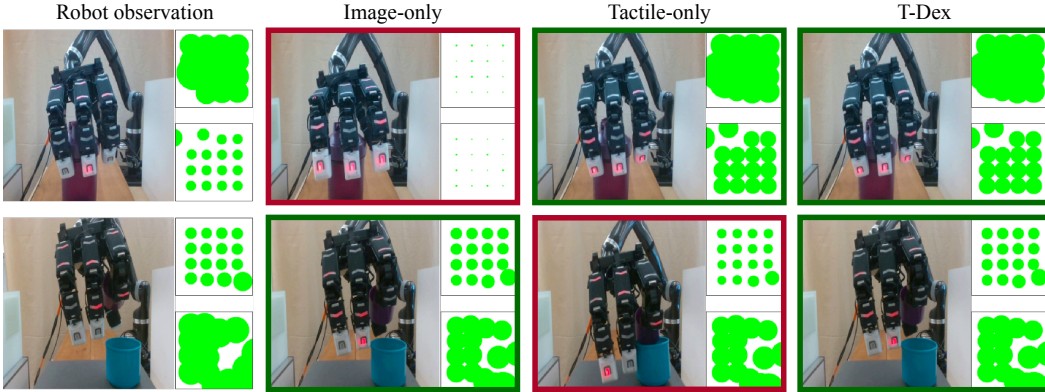

Figure 4: Visualization of the camera image, top-two activated tactile sensors, and their nearest neighbors for T-DEX and baselines. We observe that baselines fail to either recognize the contact with the object (image-only) or to capture the position of the robot (tactile-only).

5. *NN-Task*: Instead of training the tactile encoder on the play data, we train it on the 6 task-specific demonstrations. Representation retrieval is same as T-DEX.

6. *Behavior Transformer (BeT) [31]*: Transformer based behavior cloning baseline that focuses on capturing distributionally multi-modal behaviors from a diverse set of demonstration data.

7. *Implicit Behavioral Cloning (IBC) [30]*: We fit an energy based model (EBM) on the joint observation and action space and choose actions that minimize the output of this EBM.

8. *Gaussian Mixture Models [53] on top of BC (BC-GMM) [32]*: A probabilistic model that represents the action space as a combination of multiple Gaussian distributions given observations.

9. **T-DEX**: This is our proposed method with the tactile encoder pre-trained on all the play data followed by nearest neighbor retrieval with image and tactile features on task data.

Quantitative results can be found in Table 1, while robot rollouts and qualitative comparisons of the baselines are shown our website `tactile-dexterity.github.io`.

## 4.2 How important is tactile sensing for dexterity?

In Table 1 we report success rates of T-DEX along with baseline methods. We observe that among the nearest neighbor based methods, we find that tactile-only (NN-tactile) struggles on Book Opening and Cup Unstacking since the hand fails to localize the objects to make first contact. On the other hand, the image-only (NN-Image) struggles on Bottle Opening and Cup Unstacking as severe occlusions caused by the hand result in poor retrievals. Using torque targets (NN-Torque) instead of tactile feedback proved useful, improving over NN-Image, but did not match using tactile.

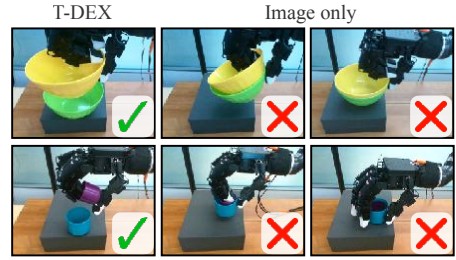

Figure 5: Visualization of the failure modes of our Image only baseline. Without tactile information, the robot applies either too much force or does not correctly contact with the objects.

We find that combining the coarse-grained localization ability of NN-Image along with the fine-grained manipulation of NN-Tactile results in the strongest results across all tasks. To further analyze why T-DEX performs so well, we visualize the nearest neighbors of states for the image-only and tactile-only methods in Figure 4. And show additional failure modes for NN-Image in Figure 5. Our method produces neighbors that seem to capture the state of the world better than other baselines. More details of the robot policy can be found in Appendix B.2 and E.

### 4.3 Does pre-training on play improve tactile representations?

To understand the importance of pre-training, we run NN-Task, which pre-trains tactile representations on task data. As seen in Table 1, this baseline does quite well on the simpler Joystick Movement task. However, on harder tasks, particularly the Unstacking tasks and Bottle Opening, we find that NN-Task struggles significantly. This can be attributed to poor representational matches when trained on limited task data. To mitigate this, we also try training the encoder with a combination of successful and failed demonstrations on the Bowl Unstacking task, getting a success rate of 30%, which shows no improvement in task performance.

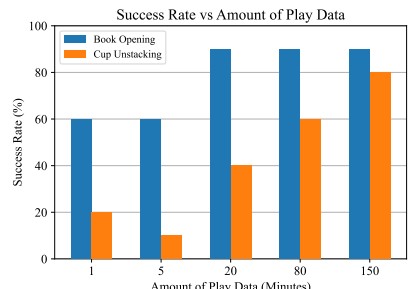

Figure 6: Success rate on unstacking tasks with varying amount of play data. Training only on task data performs moderately well, but is outperformed with just 20 minutes of play.

To provide further evidence for the usefulness of tactile pretraining, we plot the gains in performance across varying amounts of play data in Figure 6. We see that for easier tasks like Book Opening, even small amounts of play data (20 mins) is sufficient to achieve a 90% success rate. However, for harder tasks like Cup Unstacking, we see steady improvements in success rate with larger amounts of play data.

### 4.4 Analysis of the design decisions in T-DEX

**Choice of Policy Class:** T-DEX uses a non-parametric class instead of commonly used parametric ones. To study the importance of this choice, we compare across a variety of fully parametric policy architectures including BC [29], IBC [30], BeT [31], BC-GMM [32] in Table 1. We observe that parametric methods fail to learn high-quality policies and significantly overfits on the small amount of data with the high-dimensional action space. This emphasizes the need for retrieval-based policies, which have shown success with few demonstrations [35, 5].

**Choice of the encoder architecture:** A critical component in T-DEX is the architectural details in representing and processing tactile data. In this section, we examine various approaches to represent tactile features. For simplicity, we study a subset of the tasks, Book Opening and Cup Unstacking and show the results in Table 2. Each encoder is trained using BYOL on the play dataset with the same augmentations used in the main method. We compare our main encoder, AlexNet with few different architectures: (a) *ResNet:* A standard Resnet-18 [46] with weights pre-trained on the ImageNet [47] classification task. (b) *3-layer CNN:* A CNN with three layers initialized with random weights. (c) *Stacked CNN:* Rather than laying out the sensor data of the fingers spatially in the image, we consider stacking the sensor output into one 45-channel image. (d) *Shared CNN:* We pass individual pad values to the same network and concatenate the outputs. (e) *Raw Tactile:* We flatten the raw tactile data into a 720-dimensional vector instead of using an encoder.

Table 2: Success rates of various representations for tactile data on two of our tasks.

|  | T-DEX | ResNet | 3-layer | Stacked | Shared | Raw |
|---|---|---|---|---|---|---|
| Book Opening | **9/10** | **9/10** | 6/10 | 5/10 | 2/10 | 5/10 |
| Cup Unstacking | **8/10** | 6/10 | 3/10 | 1/10 | 1/10 | 3/10 |

We find that both T-Dex and ResNet perform similarly on Book Opening, although ResNet takes significantly more computation for the same results. On Cup Unstacking we find that ResNet performs a little worse than T-Dex, which further informs our architectural choice. While, one may conclude that smaller architectures are better, we see that a simpler 3-layered CNN also performs poorly and does not reach the performance of either of the larger models.

**Structure of the tactile input:** Apart from the architecture, we find that the structure of inputting tactile data from individual tactile pads is also important. For example, we find that stacking tactile pads channel-wise is substantially worse than T-Dex that stacks the tactile pads spatially. Similarly we find that using a shared encoder for each tactile pad is also poor. This is perhaps because of the noise that exists in high-dimensional raw tactile data, which is difficult to filter out with the stacked and shared encoders. Hence, one spurious reading in an unused tactile pad could yield an incorrect neighbor, producing a bad action. This hypothesis is further substantiated by the result of the Raw Tactile method, which is roughly on par with the Stacked method.

**Alternative representations:** We additionally run three experiments with different tactile representations on the Bowl Unstacking task to analyze our choice of representation. We run PCA on the Raw Tactile features on the play dataset and use the top 100 components as features, achieving a success rate of 40%. When PCA fails, it is not able to capture fine-grained tactile information that is necessary to solve the task. Next, we sum the activations of each 4x4 tactile sensor in each dimension to create a 45-dimensional feature, which does not succeed on any task. Finally, we shuffle the order of the pads in the tactile image, which achieves 20% success, which is much lower than using the structured layout (Section 3.1), showing that the layout of tactile data is crucial for performance.

### 4.5 Generalization to Unseen Objects

To examine the generalization ability of T-Dex, we run the Bowl Unstacking and Cup Unstacking tasks with unseen crockery. T-Dex policies for each of these tasks were trained using only one set of objects, seen in Figure 1 (b). Without any modifications, the policy is then run on new objects, with the objects placed in 10 different configurations. As seen in Figure 7, T-Dex is able to generalize to these new objects in varying degree. For objects of similar shapes and size as the training object, T-Dex does quite well. However, it begins to fail when the objects change significantly. For example, the inner bowl in the right-most column for Bowl Unstacking is too small for the hand to pick up.

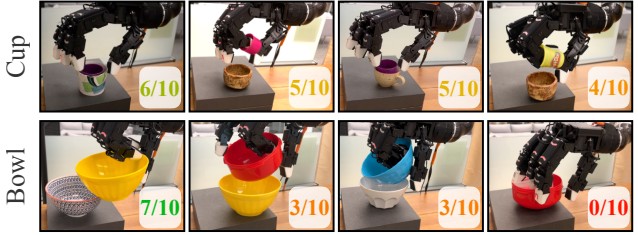

Figure 7: We show success rates of T-Dex on objects not seen during demonstration collection.

## 5 Limitations and Conclusion

In this work, we have presented an approach for tactile-based dexterity (T-Dex) that combines tactile pretraining on play data along with efficient downstream learning on a small amount of task-specific data. Our results indicate that T-Dex can significantly improve over prior approaches that use images, torque and tactile data. However, we recognize two key limitations. Although T-Dex succeeds on several out-of-distribution examples, the success rate is lower than the training setting. The second, is that our approach is currently limited to offline imitation, which limits the ability of our policies to learn from failures. Both limitations could be addressed by integrating online learning and better tactile-vision fusion algorithms. While these aspects are out of scope to this work, we hope that the ideas introduced in T-Dex can spur future work in these directions.

## Acknowledgments

We thank Vaibhav Mathur, Jeff Cui, Ilija Radosavovic, Wenzhen Yuan and Chris Paxton for valuable feedback and discussions. This work was supported by grants from Honda, Meta, Amazon, and ONR awards N00014-21-1-2758 and N00014-22-1-2773.

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

## Appendix

## A    Related Work

### A.1    Learning in Dexterous Manipulation

There are several methodologies for using learning in dexterity. Model-based reinforcement learning (RL) methods have been shown to work in both simulation [54, 55] and the real world [56, 1]. Model-free RL has been used to train policies both in simulation [57, 58] and directly on hardware [59]. Simulation to real transfer has also shown success [60, 2, 61, 52], though it often requires extensive randomization, significantly increasing training time. The use of expert demonstrations can reduce the amount of real-world interactions needed to learn a dexterous policy [62, 59, 4]. The works mentioned above either use visual observations or estimates of object state, which suffer during heavy occlusion of the object.

### A.2    Representation Learning for Robotics

Learning concise representations from high-dimensional observations is an active area of research in robotics. A wide variety of approaches using auto-encoders [63, 50, 64], physical interaction [65], dense descriptors [66], and mid-level features [67] have been studied.

In computer vision, self-supervised learning (SSL) is often used to pre-train visual features from unlabeled data, improving downstream task performance. Contrastive methods learn features by moving features of similar observations closer to one another and features of dissimilar observations farther from one another [68, 69]. These methods require sampling negative pairs of datapoints, which adds an additional layer of complexity. Non-contrastive methods typically try to learn features by making augmented versions of the same observation close [44, 70] and do not require sampling negative examples. Self-supervision has been adopted for visual RL [71, 72, 49, 50] and robotics [34, 35, 5, 36] to improve sample efficiency and asymptotic performance. SSL methods have also been applied to other sensory inputs like audio [73] and depth [74]. We build on this idea of self-supervision and extend it to tactile observations. However, unlike visual data, for which large pretrained models or Internet data exists, neither are available for tactile data. This necessitates the creation of large tactile datasets, which we generate through robot play.

## B    Experiment Details

### B.1    System Details and Robot Setup

Our robotic system, visualized in Figure 8, consists of a robotic arm and hand. The arm is a 6-dof Kinova Jaco and the hand is a 16-dof Allegro hand with four fingers. The arm can be teleoperated through the built-in Kinova joystick, while the the hand can be teleoperated using the the Holo-Dex framework [5]. Here, our teleoperator uses a virtual reality headset to both visualize robot images and control the hand in real time. The headset returns a pose estimate for each finger of the hand which is re-targeted to the Allegro Hand. Inverse Kinematics is then used to translate target Cartesian positions in space to joint

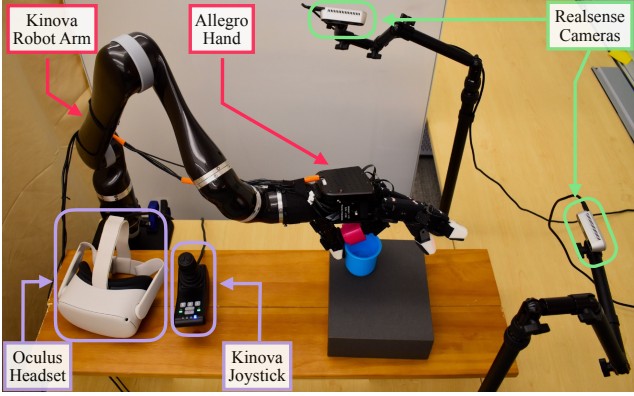

Figure 8: Hardware setting of T-DEX. We use an Oculus Headset to teleoperate the Allegro hand and the built in Kinova joystick to control the arm. Visual observations are streamed through two different Realsense cameras and tactile observations are saved with XELA touch sensors on the Allegro hand.

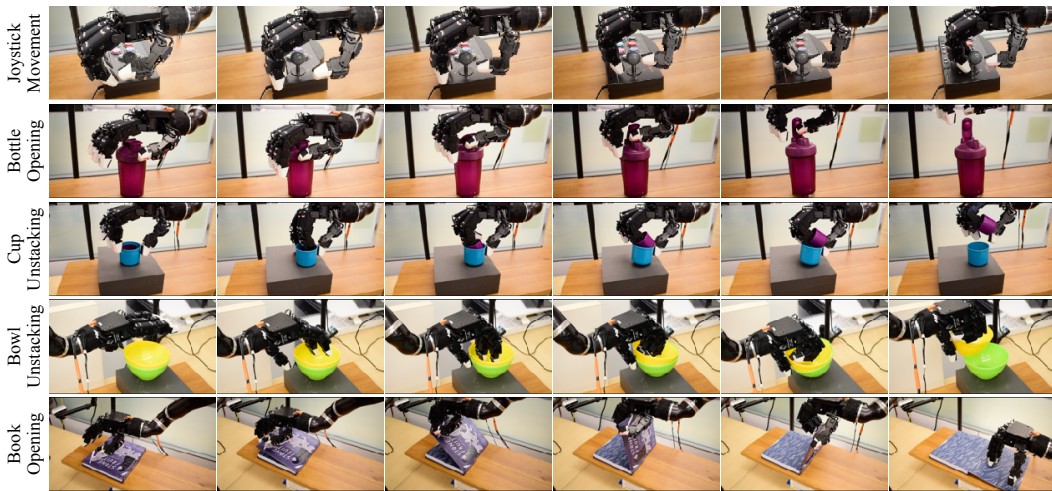

Figure 9: Visualization of all robot rollouts from T-DEX policies. Note the severe visual occlusions when the robot makes contact with the object.

angles, which are fed into the low-level hand controller. To achieve robust position control, we use a low-level PD joint position controller with gravity compensation to allow the robot to maintain a hand pose at different orientations in space. Our action space is Cartesian position and orientation of the arm (3D position and 4D quaternion for orientation) and the 16-dimensional joint state of the hand for a total of 23 dimensions.

The Allegro hand is fitted with 15 XELA uSkin tactile sensors, 4 on each finger and 3 on the thumb. Each sensor has a 4x4 resolution output of tri-axial force reading (forces in translational x, y, z) information, which amounts to a 720-dimensional tactile reading. The force readings are uncalibrated, susceptible to hysterisis, and can change when strong magnets or metals are in the vicinity. Due to this, we opt against explicit calibration of the 720 sensor units. To supplement the tactile sensors, we also use two RGB cameras with 640x480 resolution to capture visual information in the scene, though our policies only uses information from one to execute. Our choice of camera for tasks depends on which one captures the most visual information about the objects to ensure fairness when comparing to baselines and enable better joint vision and tactile control.

## B.2    Task Details

Here we explain the details of our tasks and visualize the robot rollouts for each of them in Figure 9.

1. **Joystick Movement:** Starting over an arcade gamepad, the hand is tasked with moving down and pulling a joystick backwards. This task is difficult because the hand occludes the gamepad when manipulating it. We collect demonstrations of the joystick in two different positions and evaluate on different positions and orientations not seen during training. A trial is successful if the joystick has been pulled within 60 seconds.

2. **Bottle Opening:** This task requires the hand to open the lid of a bottle. We collect three demonstrations with the bottle orientation requiring the use of the thumb, and three other requiring the use of the middle finger. The task is considered successful if the lid is open within 120 seconds.

3. **Cup Unstacking:** Given two cups stacked inside one another, the tasks is to remove the smaller cup from the inside of the larger one. In addition to occlusion, this task requires making contact both the inner and outer cups before lifting the inner cup with the index finger. It is considered a success if the smaller cup is raised outside the larger cup without dropping it or knocking the cup off the table within 240 seconds.

4. **Bowl Unstacking:** This task is similar to the previous, but with bowls instead of cups. Since the bowls are larger, multiple fingers are required to lift and stabilize them. A run is successful if it has lifted the bowl within 100 seconds.

5. **Book Opening:** This task requires opening a book with three fingers. After making contact with the cover, the hand must pull up with an arm movement, remaining in contact until it is fully open. The task is considered a success if the book is open within 300 seconds.

### B.3 Baseline Details

Here we explain the details of all our baselines. Unless explicitly explained all the image features are received by an image encoder trained with BYOL on the task-specific demonstrations and all the tactile features are received by a tactile encoder pre-trained on all the play data. Success rates of each of the baselines are shown in 1.

1. *Behavior Cloning (BC) [51]*: We train a neural network end-to-end to map from visual and tactile features to actions.

2. *Nearest Neighbors with Image only (NN-Image) [35]*: We perform nearest neighbors with the image features only. During evaluation, to ensure fairness we use viewpoints that can convey maximal information about the scene.

3. *Nearest Neighbors with Torque and Image (NN-Torque) [52]*: We perform nearest neighbors with the output torques from our PD controller and visual observations. The torque targets can be used as a proxy for force, providing some tactile information.

4. *Nearest Neighbors with Tactile only (NN-Tactile)*: Nearest neighbors with the tactile features trained on play data. Unlike T-DEX we do not use vision data for this baseline.

5. *Nearest Neighbors with Tactile Trained on Task Data (NN-Task)*: Instead of training the tactile encoder on the play data, we train it on the 6 task-specific demonstrations.

6. *Behavior Transformer (BET) [31]*: We train a transformer to predict action modes and offsets given image and tactile features. This method is known to capture the multi-modality of the environment given unlabeled multi-task data.

7. *Implicit Behavioral Cloning (IBC) [30]*: We train a model that outputs the energy of the representation space given image and tactile features and optimize for actions that minimize the energy landscape. This method is proven to capture the stochastic nature of the environments and recover better in out-of distribution modes compared to the explicit behavior cloning (BC) approach.

8. *Gaussian Mixture Models [53] on top of BC (BC-GMM)*: We fit a Gaussian Mixture Model on the action space and sample actions from a weighted sum of Gaussian component densities from given image and tactile features.

9. *Nearest Neighbors with Tactile Trained on Play Data (T-DEX)*: This is our main method with the tactile encoder pre-trained on all the play data followed by nearest neighbor retrieval on task data.

For simplicity, we secure the joystick, bottle, and book to the table. This mimics having another manipulator keep the object in place while the hand manipulates the object. The bowls and cups are not secured, making the problem of unstacking much more difficult.

To ensure fair evaluation, we start each method with the object in the same configuration for each index trial. This corresponds to 10 different starting positions, each of which is used at the start of each baseline run.

## C Tactile Data Details

In this section we give more details regarding the mapping of the tactile readings to our robotic setup and the importance of each of the sensor readings to our tasks.

## C.1 Mapping of the tactile images

Figure 10 showcase where each tactile pad is on the Allegro hand and how they are mapped to tactile readings and images.

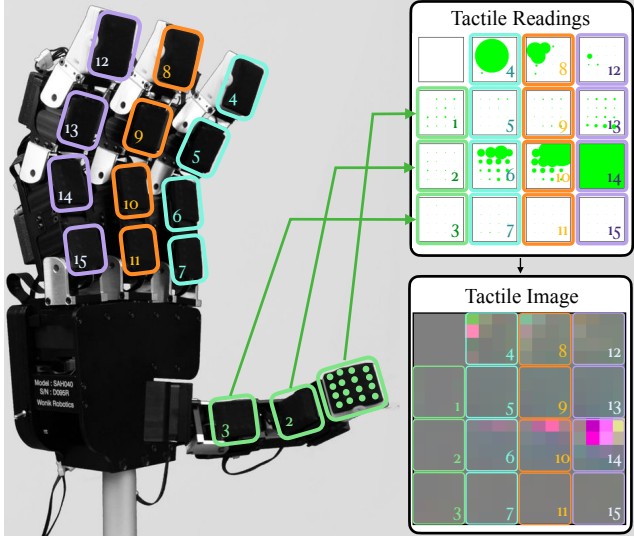

Figure 10: We show the location of each tactile pad and how they are mapped to the corresponding tactile readings and images. Locations could be observed by following the numbered pads and visualizations.

## C.2 Contribution of each tactile reading to tasks

In order to explore the contribution of each tactile reading to our tasks we created a figure to show the heatmaps for the variances of each taxel through robot trajectories for different tasks. We observe that the lower segments of the fingers have very high variance due to the lower segments touching the palm of the hand in most of the tasks. The variance in the other segments show the importance of each pad as follows:

- **Bowl Unstacking**: The 2nd and 3rd segment tactile pads of the thumb.
- **Bottle Cap Opening**: Tips of middle, index and the thumb.
- **Cup Unstacking**: Tip, 2nd and 3rd segment of the thumb.
- **Joystick Movement**: Tips of index, middle and ring finger.
- **Book Opening**: Tips of index, middle and ring finger and the 3rd segment of the thumb.

Heatmaps for the variances can be seen in Figure 11.

## D    Model Details

Here we provide additional details about our method and baselines for easier reproduction.

For all image-based models, we normalize the inputs based on the mean and standard deviation of the data seen during training. For the tactile-based models, we normalize the inputs to be within the range $[0, 1]$.

### D.1    BYOL Details

**Bootstrap your own Latent:**    BYOL has both a primary encoder $f_\theta$, and a target encoder $f_\xi$, which is an exponential moving average of the primary. Two augmented views of the same observation $o$ and $o'$ are fed into each to produce representations $y$ and $y'$, which are passed through projectors $g_\theta$

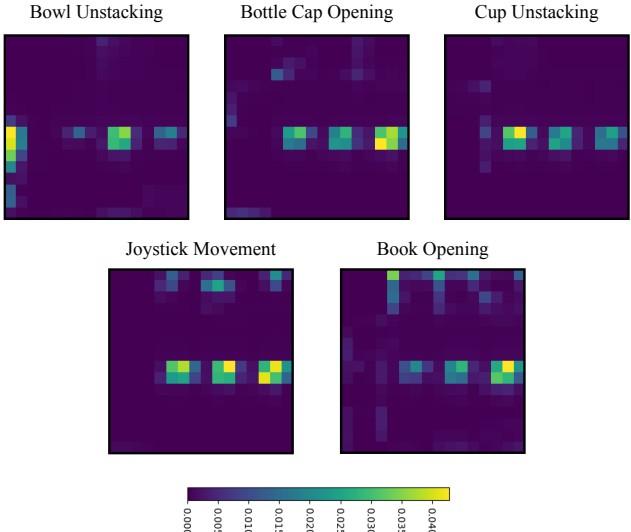

Figure 11: Heatmaps of the variances of each taxels through all of our robot trajectories.

and $g_\xi$ to produce $z$ and $z'$, which are higher dimensional. The primary encoder and projector are then tasked with predicting the output of the target projector. After training, we use $f_\theta$ to extract features from observations.

**Using Tactile Readings with BYOL:** After getting the tactile images as mentioned in 3.1, we scale the tactile image up to 224x224 to work with standard image encoders. For the majority of our experiments, we use the AlexNet [45] architecture, also starting with pre-trained weights. Unlike SSL techniques in vision [44], we only apply the Gaussian blur and small random resized crop augmentations, since other augmentations such as color jitter and grayscale would violate the assumption that augmentations do not change the tactile signal significantly. Importantly, unlike vision, since all of the tactile data is collected in the frame of the hand, the sensor readings are invariant to hand pose and can be easily reused between tasks.

The complete list of BYOL hyperparameters has been provided in Table 3. We take the model with the lowest training loss out of all the epochs.

### D.2 Nearest Neighbors Details

We give equal weight to visual and tactile distances for all of the tasks except bottle cap, where tactile and image features were given weights of 2 and 1, respectively. We do this because the quality of the neighbors on image data was poor and emphasizing the tactile data slightly vastly improves performance.

While executing NN imitation, we keep a buffer of recently executed neighbors that we call the reject buffer. Given a new observation, we pick the first nearest neighbor not in the reject buffer. This prevents the policy from getting stuck in loops if a chain of neighbors and actions are cyclical. We set the reject buffer size to 10 for every task except Joystick Pulling, which is set to 3. The buffer, combined with the 2cm spatial subsampling are critical for the success of NN policies.

### D.3 BC Details

We train BC end-to-end using standard MSE loss on the actions with the same learning rate as BYOL and a batch size of 64.

| Parameter | Value |
| --- | --- |
| Optimizer | Adam |
| Learning rate | $1e^{-3}$ |
| Weight decay | $1e^{-5}$ |
| Max epochs | 1000 |
| Batch size (Tactile) | 1024 |
| Batch size (Image) | 64 |
| Aug. (Tactile) | Gaussian Blur (3x3) (1.0, 2.0) $p = 0.5$ |
| | Random Resize Crop (0.9, 1.0) $p = 0.5$ |
| Aug. (Image) | Color Jitter (0.8, 0.8, 0.8, 0.2) $p = 0.2$ |
| | Gaussian Blur (3x3) (1.0, 2.0) $p = 0.2$ |
| | Random Grayscale $p = 0.2$ |

Table 3: BYOL Hyperparameters.

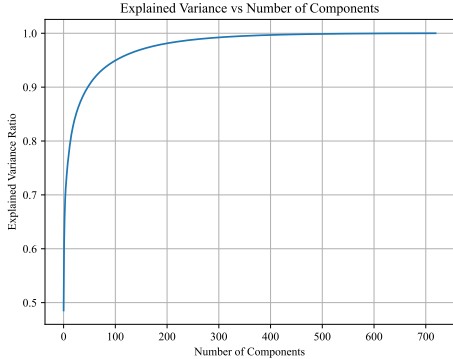

Figure 12: Explained variance ratio for PCA on the play tactile data. Most variance is captured in the first 100 components.

### D.4 NN-Torque Details

Our hand does not have torque sensors, but is actuated by torque targets from a low-level PD position controller. We use the torque targets as a proxy for torque information since the desired torque will be higher when the finger is in contact with an object, but trying to move further inside, and lower when it is not in contact.

### D.5 PCA Details

We run PCA on the tactile play data and take the top 100 components for use as features. The captured variance is about 95% and the entire explained variance ratio can be seen in Figure 12. By visualizing the reconstructions (Figure 13), we can see that it retains a majority of the tactile information.

### D.6 Shuffled Pad Details

For this experiment, we permute the position of the 15 4x4 tactile sensors using the same permutation for both pretraining and deployment. This ensures that we're inputting the same data from each sensor to each location in the tactile image, but does not leverage the spatial locations of the pads on

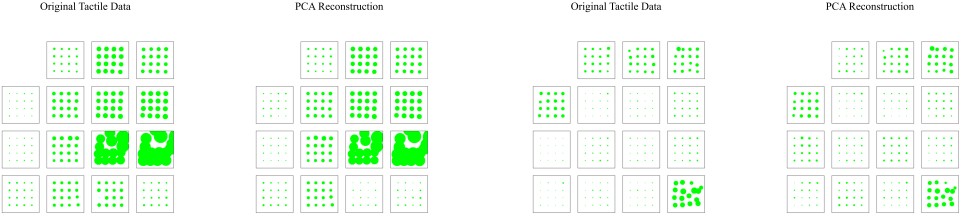

Figure 13: Tactile data and the PCA reconstruction of two using 100 components for two tactile readings. Most of the information is preserved, but we can see minor differences in magnitude and offset.

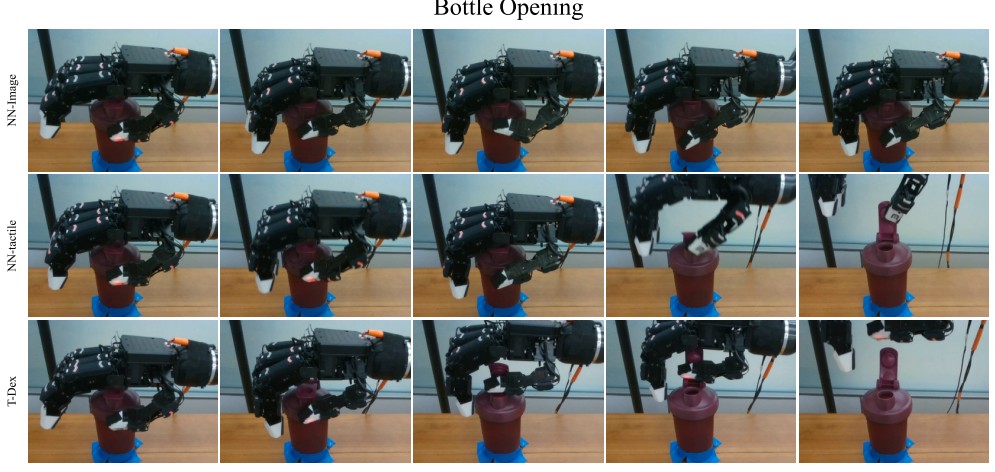

Figure 14: Additional rollouts for the Bottle Opening task.

the hand. If spatial layout had no effect, we would expect no difference in the performance between this and T-DEX.

## E   Additional Rollouts

We visualize extra rollouts for each task in Figures 14-18.

## F   Tactile Image Visualization

We visualize tactile images for each task in Figures 19-23.

Book Opening

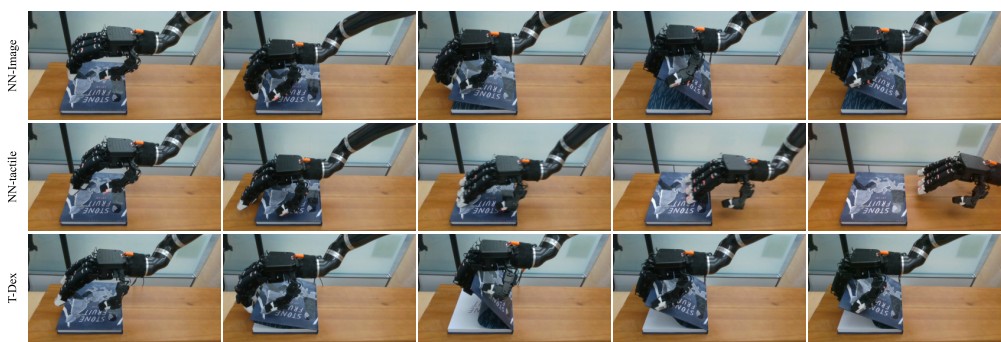

Figure 15: Additional rollouts for the Book Opening task.

Bowl Unstacking

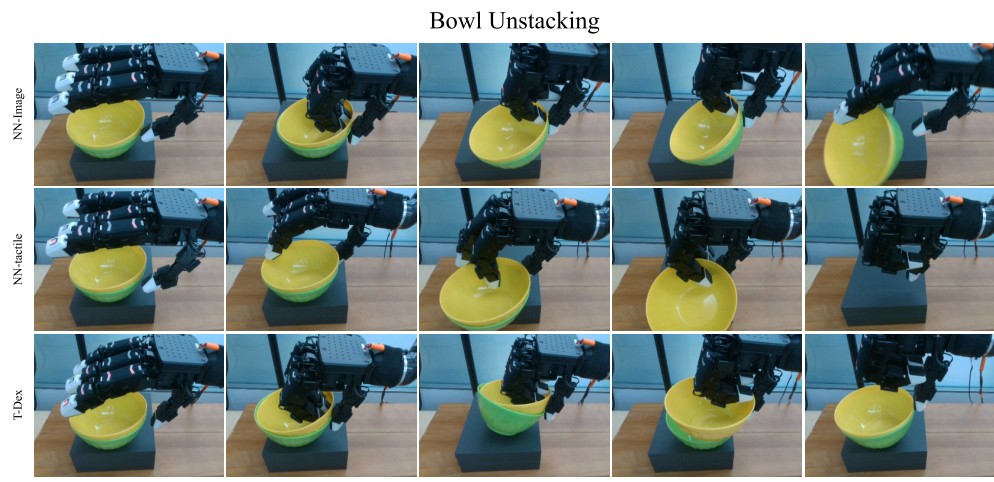

Figure 16: Additional rollouts for the Bowl Unstacking task.

Cup Unstacking

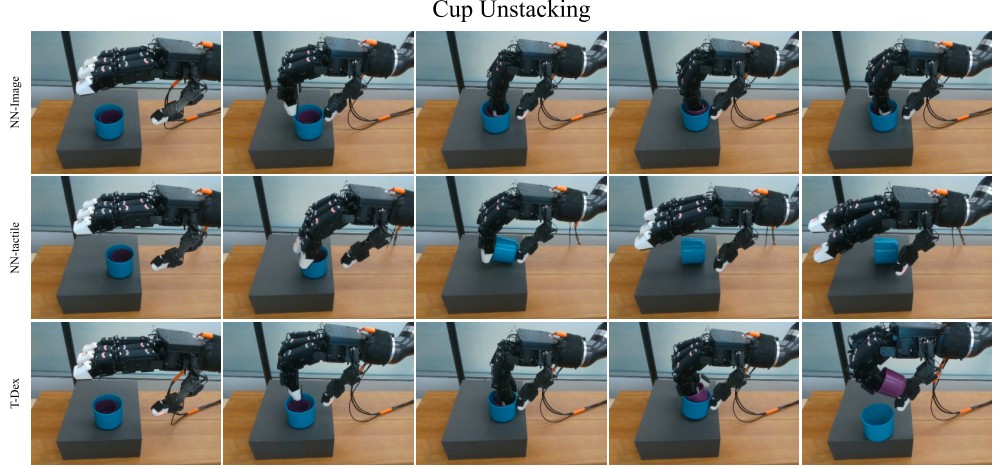

Figure 17: Additional rollouts for the Cup Unstacking task.

Joystick Pulling

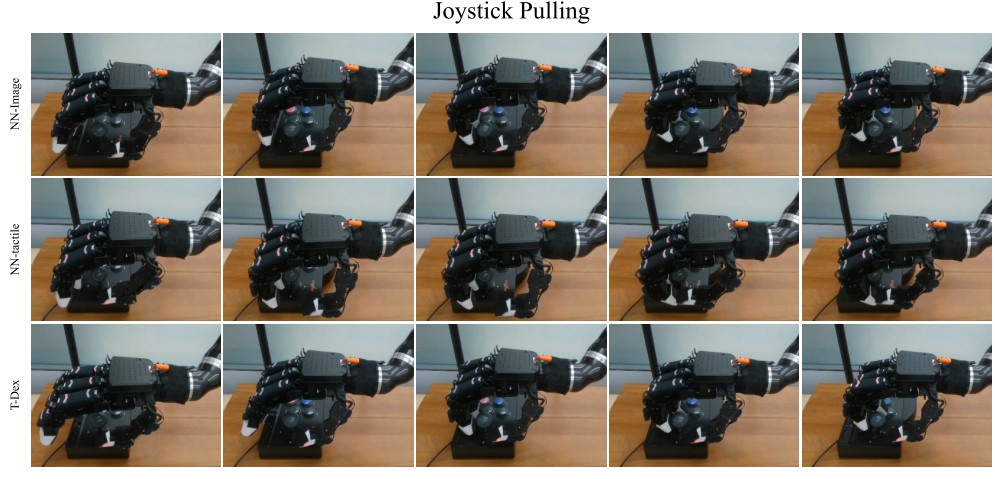

Figure 18: Additional rollouts for the Joystick Pulling task.

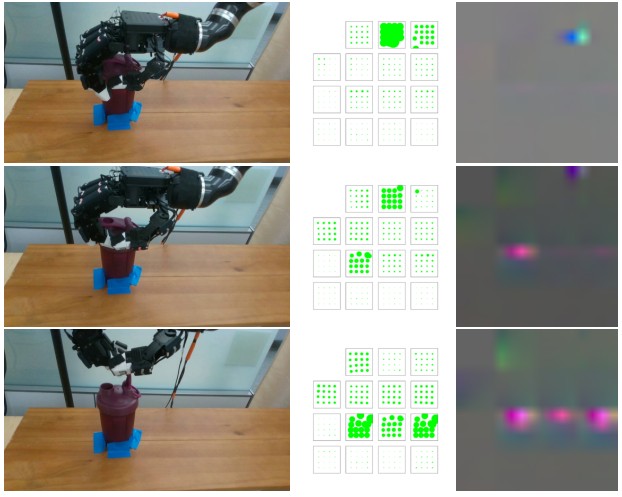

Figure 19: Tactile Image for the Bottle Opening task.

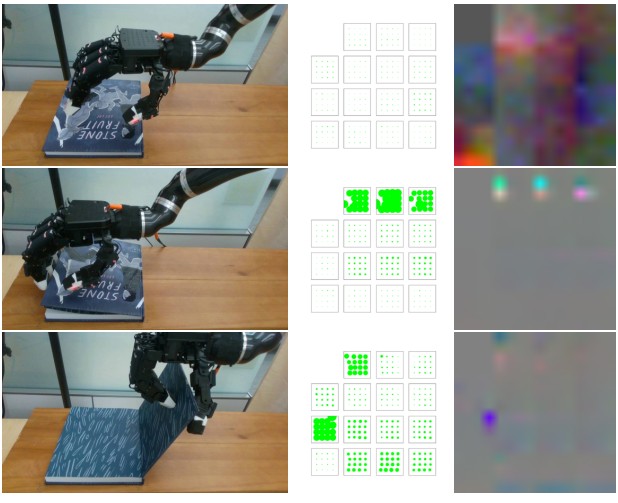

Figure 20: Tactile Image for the Book Opening task.

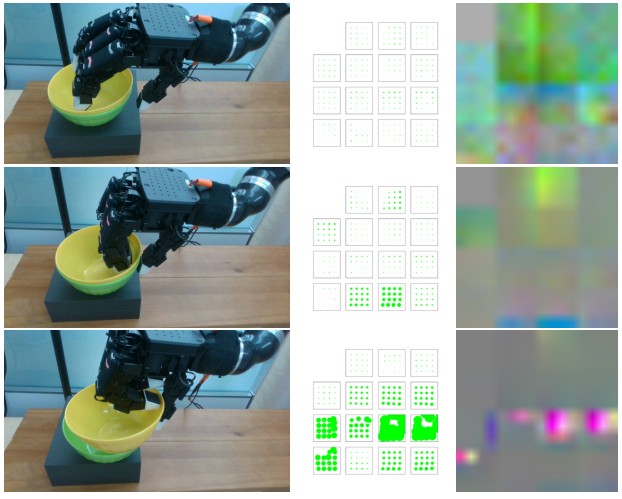

Figure 21: Tactile Image for the Bowl Unstacking task.

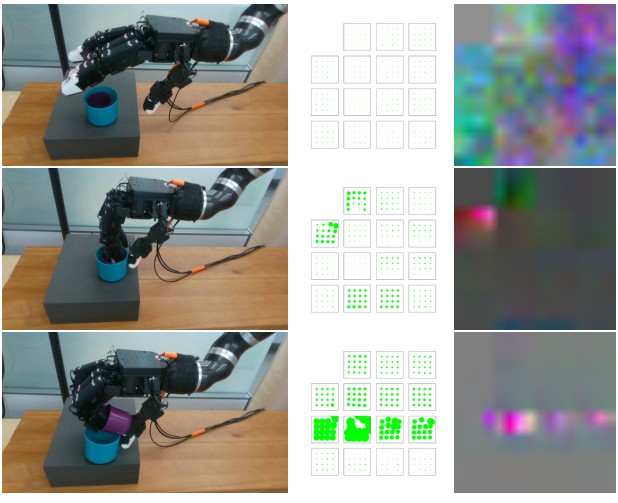

Figure 22: Tactile Image for the Cup Unstacking task.

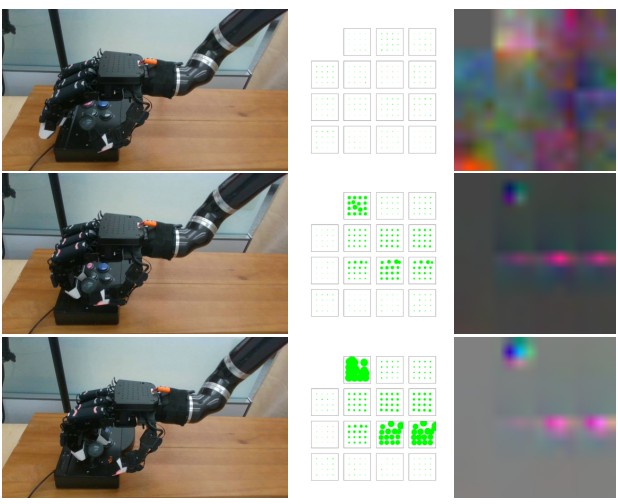

Figure 23: Tactile Image for the Joystick Pulling task.

