# OpenReview forum: "Dexterity from Touch: Self-Supervised Pre-Training of Tactile Representations with Robotic Play"
_robot-learning.org/CoRL/2023/Conference — CoRL 2023 Poster_

### Official Review · Reviewer_27e9 · 2023-06-26

**Confidence:** 5
**Originality:** Very Good
**Technical Quality:** Very Good
**Clarity Of Presentation:** Excellent
**Impact:** 4

**Recommendation:**

Strong Accept: I recommend accepting the paper and will argue for my recommendation even if other reviewers hold a different opinion.

**Review:**

## Strengths
* Well motivated technique for pretraining tactile representations from unstructured play data. The techniques ability to learn from either successful or unsuccessful demonstrations seems particularly well aligned with the realities of teleoperation for dexterous manipulation studied in this work, where the failure rate of teleoperators is 5/ 6.
* Clear description of the method and helpful and practical descriptions of how the dataset was constructed and filtered.
Strong evidence under extensive real world evaluations (50 hours) of the efficacy of the proposed technique.
* Good ablations quantifying design decisions and highlighting the importance of pre-training on play, and how separate tasks scale differently with more play.
* (Pending the actual release) The 2.5 hours of high rate vision and tactile sensing play data could be useful to the community as a source of pretraining or for studying multimodal representation learning.


**Quality Of The Limitations Section:**

Limitations are addressed clearly

**Questions For Rebuttal:**

N/A

**Robotics Focus:**

Sufficient demonstration on hardware

**Summary Of Paper:**

This work presents a method for learning self-supervised tactile encoders from unlabeled play data that are useful in downstream dexterous manipulation tasks.

**Summary Of Recommendation:**

This work shows evidence that tactile representations learned from large amounts of task-agnostic play data are useful in downstream dexterous imitation learning tasks, and generalize to new visually distinct object instances. This feels like an important finding to communicate to the community, as it derisks a strategy for learning from more scalable forms of supervision than those currently considered for tactile dexterous manipulation.

---

> ### Author Response · Authors · 2023-08-08
> **Author Response to Reviewer 27e9**
>
> We greatly appreciate the comments mentioned on the review. The play data will be publicly released with additional sensing modalities (2 RGBD cameras and full tactile). We hope that this data is useful for the community to build on multi-sensory representations.
>
> We are glad to see the appreciation of the contributions of our paper, thank you. If any additional concerns arise, we will be happy to address them!
>
> Regards,
>
> Authors of Paper 49.

---

### Official Review · Reviewer_rRsP · 2023-06-29

**Confidence:** 4
**Originality:** Good
**Technical Quality:** Very Good
**Clarity Of Presentation:** Very Good
**Impact:** 3

**Recommendation:**

Weak Accept: I recommend accepting the paper, but will not argue for my recommendation if the majority of other reviewers have a different opinion.

**Review:**

Strenghs
- The paper is written clearly.
- Imitation learning dextrous manipulation, unsupervised learning tactile representation, leveraging play data, and pre-training for learning manipulation are significant issues that the paper tackles.
- The description of the method and experiments sound technical.
- The originality of the paper is to address the pre-training unsupervised tactile representation with play data for learning downstream dexterous manipulation.

Weaknesses
- The downstream tasks, such as opening a book, bottle cap opening, and unstacking cups, do not seem challenging. A simple parallel gripper can execute these tasks. The tasks do not express the dexterity of the multi-finger robot hand.
- Their nearest neighbor method does not have scalability. Their method may be used for more complex manipulation task which requires continuous control.

**Quality Of The Limitations Section:**

Limitations are addressed clearly

**Questions For Rebuttal:**

- Listing the contributions of the paper in the introduction may help the readers to understand these contributions easily.
- The experiment section should show the five downstream tasks again and introduce the teleoperation system.
- If the proposed method can be used for the non-nearest neighbor method, explaining how to do that helps the reader to understand the significance of the paper.

**Robotics Focus:**

Sufficient demonstration on hardware

**Summary Of Paper:**

This paper presents a self-supervised pre-training learning method for tactile encoders using play data.
This paper uses Holo-Dex framework as a teleoperation framework, Allgro hand as a robot hand, Kinova robot arm, Realsense cameras, and XELA uSkin as a tactile sensor.
In the first phase, this method collects play data and learns tactile representation.
In the second phase, the method collects expert task demonstrations with vision and learns tasks using the nearest neighbor method to select a hand and finger pose.

The main contribution of the paper is to present a method for self-supervised pre-training of tactile presentation with play data for learning dextrous manipulation.

**Summary Of Recommendation:**

The paper is well-written and has clarity, originality, and significance.
However, the downstream tasks do not seem challenging dexterous manipulation requiring multiple fingers, and it is not unclear whether the proposed method can be used for more complex manipulation.
Therefore, the reviewer recommends weak accept.

---

> ### Author Response · Authors · 2023-08-08
> **Author Response to Reviewer rRsP**
>
> We thank the reviewer for their constructive feedback and appreciate the concerns and the suggestions brought up. We'll address each rebuttal question / points made separately in the below.
>
> **Listing the contributions of the introduction**
> > Listing the contributions of the paper in the introduction may help the readers to understand these contributions easily.
>
> Currently, we have listed the contributions of the paper at the end of the Introduction section, between lines 60 to 65. Here we describe the main contributions of the paper as:
>
> 1. T-DEX improves upon vision-only and torque-only imitation models with over a 170% improvement in average success rate.
> 2. Play data significantly improves tactile-based imitation, with an average of 58% improvement over tactile models that do not use play data.
> 3. Ablations on different tactile representations and architectures show that the design decisions in T-DEX are important for high performance.
>
> We hope this list is sufficient. If not, we are happy to add additional explanations.
>
> **Experiment section**
> > The experiment section should show the five downstream tasks again and introduce the teleoperation system.
>
> This is a good point! We initially added the task figure and the teleoperation system in Section 3. However due to the page limit constraints we moved it to Appendix B of the supplementary to give us space to focus on the analysis of the design choices we had such as:
> * Different modalities used in nearest neighbors.
> * Importance of amount of play-data used and improvement on the downstream tasks.
> * Architecture, structure of the tactile input and alternative modalities.
> * Generalization capabilities of our approach.
>
> We believed that these experiments play a more important role on the main contributions which largely relies on usage and structure of tactile data, hence, wanted to provide more detailed information and figures to support these bullet points. We hope that we get an extra page for the camera ready version of the paper so we can move this figure back into the main paper.
>
> **Simpleness of the downstream tasks**
> > The downstream tasks do not seem challenging dexterous manipulation requiring multiple fingers.
>
> Thank you for bringing up this concern! We agree that some of our easier tasks like book opening and bottle cap opening can be done with a parallel jaw gripper. However, the other tasks require additional degrees of freedom. For example, the unstacking tasks require the fingers to bend to scoop out the inner object. Furthermore, we would like to note that these tasks are also quite hard for humans to teleoperate, with a success rate of only 16%.
>
> **Scalability and Non nearest-neighbor methods**
> > Their nearest neighbor method does not have scalability. Their method may be used for more complex manipulation task which requires continuous control.
> > If the proposed method can be used for the non-nearest neighbor method, explaining how to do that helps the reader to understand the significance of the paper.
>
> We use the nearest-neighbor approach due to poor training parametric policies with few demonstrations. As seen from our fully parametric baselines, learning with only six demonstrations leads to significant overfitting and failure of policies during deployment. For full transparency, we also add this limitation of scalability in the limitations section (Section 5).
>
> We believe that leveraging tactile information using encoders trained on task-agnostic play data will be useful for parametric approaches as well. However, this would need a much larger number of demonstrations for downstream tasks or using online imitation learning algorithms. We believe that this could give more scalable policies and address the overfitting issue in parametric models. We are currently actively investigating both directions.
>
>
> Thank you again for your feedback. We hope that our responses and additional discussion have addressed your concerns. We are happy to address any other concerns that you may have!
>
>
> Regards,
>
> Authors of Paper 49.

---

> > ### Comment · Reviewer_rRsP · 2023-08-08
> > **Contributions**
> >
> > The following list is a summary of results. This is not contribution. As contributions, contributions to the academic field of robotics are written.
> >
> > > 1. T-DEX improves upon vision-only and torque-only imitation models with over a 170% improvement in average success rate.
> > > 1. Play data significantly improves tactile-based imitation, with an average of 58% improvement over tactile models that do not use play data.
> > > 1. Ablations on different tactile representations and architectures show that the design decisions in T-DEX are important for high performance.

---

> > > ### Author Response · Authors · 2023-08-08
> > > **Respond for Contributions**
> > >
> > > We apologize for the misunderstanding, that is correct!
> > > We would like to summarize the novel contributions of this work as:
> > > 1. Creating a robotic system to combine vision and task-agnostic tactile representations for multi-fingered dexterous manipulation.
> > > 2. Demonstrating that useful tactile encoders can be trained via self-supervision on play data.
> > > 3. Experimentally showing that this tactile information greatly improves downstream performance on difficult, contact-rich tasks.
> > >
> > > These will be either included with the results listed or will be added separately to the Introduction section.
> > >
> > > We hope this addresses your concern. If not, please let us know and we will be happy to discuss more.

---

> ### Author Response · Authors · 2023-08-11
> **Regarding More Concerns / Questions**
>
> We greatly appreciate the time you have taken to review our paper. We would be grateful if you could inform us of any  concerns that need to be addressed in order to enhance the paper and strengthen the recommendation for acceptance.

---

### Official Review · Reviewer_Tv7A · 2023-07-16

**Confidence:** 5
**Originality:** Good
**Technical Quality:** Excellent
**Clarity Of Presentation:** Excellent
**Impact:** 4

**Recommendation:**

Strong Accept: I recommend accepting the paper and will argue for my recommendation even if other reviewers hold a different opinion.

**Review:**

### Strengths:
1.	The proposed method to pretrained tactile representations on play data performs very well in combination with visual representations. The efficacy of the tactile data is demonstrated with a rigorous comparison with non-tactile image-only observations which doesn’t perform as well for the given tasks. Although inspired by prior works using BYOL, the results in the context of using tactile data and the associated analysis are new and useful.
2.	The evaluation conducted is comprehensive and covers a wide range of baselines spanning parametric as well as non-parametric approaches. For the tasks considered, the parametric approaches overfit when provided with limited demonstrations. In this context, the proposed NN based approach adopted by T-Dex serves very useful in generalizing with sparse training data. Amongst the non-parametric methods, image-only and tactile-only are shown to perform worse than T-Dex as expected.
3.	The analysis of design choices in section 4.4 is an added bonus and the authors have done a great job testing out each component of their proposed method including architecture and tactile input structuring.
4.	The paper is very well written and is a joy to read. Each section is precise and thorough in its formulation.


### Weaknesses:
1.	Evaluating on more tasks would have added greater insight and impact to the results. Given that the play data was not dependent on successful interactions and took only 2.5 hours for 5 tasks, it seems to be relatively easy to scale to more diverse tasks.
2.	Section describing tactile pad structure is unclear. Refer to rebuttal question.

**Quality Of The Limitations Section:**

Limitations are addressed clearly

**Questions For Rebuttal:**

1. The section describing the tactile sensor data in Section 3.1 under “Feature Learning” is unclear. Perhaps, it would be helpful to refer to the original tactile data in Fig. 11 in this section. I recommend instead adding a separate figure to explain this. This can include a full view photo of the hand with all sensors along with the 16x16 tactile map with each pad numbered by the corresponding finger and axis. This will make things very clear without having to explain too much in words. I suggest adding such a figure in the supplementary.

2. What is the contribution of different tactile pads to different tasks? Did you conduct such a study? It will be interesting to see which fingers and which parts of the hand are the most impactful for performing different manipulation skills.

3. What are the limitations of non-parametric approaches compared to parametric ones? Are certain tasks more difficult in the non-parametric case, such as more complex long-horizon tasks, or ones including repetitive motions?

**Robotics Focus:**

Sufficient demonstration on hardware

**Summary Of Paper:**

The paper presents a method to learn dexterous manipulation from visuo-tactile data combining techniques of play and offline non-parametric imitation learning. To do this, the authors present a two-phase approach. In stage 1, large amounts of unrestrained play data is collected using teleoperation on a set of objects. Then an encoder is trained on the tactile sensory images using BYOL in a self-supervised fashion. In stage 2, a small number of demonstrations are collected for each task and embedded using the pretrained encoder in stage 1. During test time, the input observations are embedded and actions are outputted based on a nearest-neighbor search on the demonstration embeddings. The results indicate that the play data for tactile representation learning plays a crucial role when combined with visual data for learning dexterous manipulation.

**Summary Of Recommendation:**

The paper demonstrated how tactile play data can be leveraged for pretraining encoders for learning tactile representations using SSL to be later used for offline IL. The evaluation is thorough and the results demonstrate the efficacy of visuo-tactile learning in high-dimensional action spaces.

### Post Rebuttal
The authors have updated the paper according to the provided suggestions and added necessary details about the tactile hardware and observation space. I retain my recommendation of strong accept.

---

### Official Review · Reviewer_x3zE · 2023-07-18

**Confidence:** 4
**Originality:** Fair
**Technical Quality:** Good
**Clarity Of Presentation:** Good
**Impact:** 3

**Recommendation:**

Strong Reject: I recommend rejecting the paper and will argue for my recommendation even if other reviewers hold a different opinion.

**Review:**

While the paper is technically complete in demonstrating the idea, the main issue is probably the innovation behind, which is lacking. It is widely accepted that adding tactile data to visual information helps and improves grasping performances, which is proved again in this work. The involvement of high-dimensional tactile data in this case is no different at all. The robot play data is also well-adopted method in the current literature, although in this particular case it seems it is not exactly the robot playing by itself, but teleoperated by a human operator to play with the robot system during playing. It is unclear what the T-Dex fully represents, the dataset itself, the algorithms developed, or both. Further breakdown statistics of the tasks or objects performed during robot play helps the readers to better understand the nature of this dataset itself. The learning part is also pretty standard and well-explored using Nearest-Neighbor retrieval.

The three research questions need further improvement. Question 1 is already explored in the currently literature extensively, and proved again in this work. Question 2 is essentially a known conclusion that pre-training improves learning performances, whether it is robot play data or not. And for Question 3, it is interesting to see that a much simpler Alex-net outperforms the other more complexed algorithms, especially after pre-training. While the authors did a lot of experiments with variations on the task and objects, it would be great to focus on a few representative ones with more tests than just 10 trails for a more robust conclusion, given that the authors have already conducted a lot of experiments.

And finally, the most concerning part is the close resemblance of this paper with another submission, No. 161 See-to-touch paper for CoRL. These two papers used exactly the same setup and share almost the same paper structure (different placement of related works section). Although they are positioned differently, I don't think this should be encouraged, making the work more like a production from paper mill or automatic generation instead of genuine research.

**Quality Of The Limitations Section:**

Limitations are not well addressed

**Questions For Rebuttal:**

1. Please further elaborate on the data collected for training and the robot play part, whether it is through human teleoperation or autonomous/preprogrammed exploration of the robot by itself. Human preference could have an impact on the quality of data collected. What are the nature of the objects or tasks? Are they standard available for other researchers? What about the tactile data collected?

2. Please improve the literature review section on the current literature of tactile sensing supporting vision data for robot learning. Please improve on the current literature on high-dimensional tactile data collection and dataset. Please improve on the existing work of robot manipulation explore the scene with or without human demonstration for collection of large dataset.

3. For the results reported in Table 1, I'm surprised to see that methods such as BC, BET, IBC, and GMM failed in all cases. How are they implemented? This result is not very consistent with the existing literature and I'm not sure the way the authors implemented these methods are suitable for the proposed robot setup. Please justify how to compare these method using the authors robot platform in a fair and reasonable way for analysis.

4. The tactile data presented in Fig. 4 is not very informative for understanding. What's the meaning of the red and green boxes? How to understand the relationship of these images in what sequence? The tactile data seems to be all active in large regions without further differentiation. Please improve this figure for a clearer understanding.

5. The results reported in Fig. 5 and 7 shares the similar problem as they are not particular helpful in understanding the contribution of this experiment or this work. The results in Table 2 shares the same problem as in Table 1. Please explain and elaborate on the results and the suitability of the networks used.

**Robotics Focus:**

Sufficient demonstration on hardware

**Summary Of Paper:**

This paper presents the T-Dex framework trained with robot play data to learn non-parametric policies that combine the tactile observations with visual ones. The robot play data provides the necessary demonstration that involves high-dimensional tactile data for training skills for manipulation, which the authors achieved through the T-Dex framework in a series of tasks. By comparing with a series of existing methods, the author show that the proposed method is superior in performance in five selected tasks of object manipulation. The authors explored three research questions in experiments and generalized the proposed method to unseen objects. Limitations of this work is also explained briefly and the video demonstration is helpful to understand this work.

**Summary Of Recommendation:**

Overall, I do not recommend this paper for acceptance due to the lack of innovation with repetition of known understandings, limitations in the method and data that is not clear on sharability and reproducibility, and the close resemblance to another submission to CoRL 2023.

---

> ### Author Response · Authors · 2023-08-08
> **Author Response to Reviewer x3zE**
>
> We thank the reviewer for their constructive feedback and appreciate the suggestions to improve this paper. At the same time, we believe that there might be some misunderstandings regarding this work, which we will aim to address in this rebuttal.
>
> **Regarding novelty of T-Dex**:
> >  It is widely accepted that adding tactile data to visual information helps and improves grasping performances, which is proved again in this work.
>
> The key novelty of this work is to show that full-hand tactile and vision can be combined for dexterous, multifingered manipulation. This required creating play datasets, algorithms for learning features from play, combining visual and tactile information, and finally evaluating the system on multi-fingered dexterity. We agree that prior work has demonstrated the usefulness of tactile and vision for grasping. However, we believe there is a significant difference between the tasks of grasping (an important problem in its own right) and dexterous manipulation. Further, reviewers 27e9 and Tv7A seem to agree that this work presents novel insights in the use of tactile and vision for dexterous manipulation. To highlight the difference between grasping and multifingered dexterity, we will add additional motivation in the introduction section.
>
> **More elaboration on the play data**:
> > Please further elaborate on the data collected for training and the robot play part
>
> Here are the properties of the play data you have requested:
>
> 1. **Human guidance / prior task information**: Play data collection is done by a human teleoperator without any previous information on the downstream tasks. Play data was collected without specific structures or task prioritization, but is still highly informative due to human preference
> 2. **Tactile data collected**: During the play data collection, tactile data that is 15x16x3 in dimensionality is recorded alongside visual data. The tactile data dimensions are the same as used for downstream learning.
> 3. **Standards of the data**: All of our recorded play data (including multi-view RGBD and tactile) will be publicly released for use by other researchers. The objects used in the dataset include objects from the YCB dataset and other novelty objects from our lab.
>
> **BC, BET, IBC, GMM Baseline comparisons**:
> >  I'm surprised to see that methods such as BC, BET, IBC, and GMM failed in all cases.
>
> We have provided detailed explanation for this concern in *Comparing non-parametric approaches with parametric ones* section of the Global Response. Please refer there for explanation of this concern.
>
> **Explanation of Figures 4,5,7 and Table 2**: We appreciate all the feedback given for the figure. Unfortunately we had to cut significant amounts of explanation for the CoRL paper limit. Here is the detailed explanation you have requested:
>
> * **Figure 4**: Each column represents the nearest neighbors for each baseline given the robot observations shown on the left-most column. The Red boxes demonstrates qualitatively poor neighbors that lead to failure and the Green boxes demonstrate a good neighbor.
> * **Figure 5**: Each box represents the end observations of different policy runs.
> The main motivation of both of these figures are to convey the importance of combining both vision and tactile features, as using only one or the other may result in mis-estimating the state of the world.
> * **Figure 7**: We believe that generalization capability of a task is an important aspect of evaluating policy success. We observe that integrating tactile information to this policy enables neighbors to stay in-distribution even for the visually very different objects and give successful rollouts.
> * **Table 2**: This table explores different ways to structure and utilize the tactile information. ResNet and 3-layer baselines are there to discuss the complexity of the model. Stacked CNN explores the spatial information required between the tactile sensors. When the sensor readings are stacked as channels, sensors that are next to each other are not convoluted together which causes the model to ignore spatial information. Shared CNN explores if the encoder learns to extract different features from different sensors. In Table 2 we show that these architectural decisions matter for a higher success rate.
>
> **Close resemblance with another submission**:
> > the most concerning part is the close resemblance of this paper with another submission, No. 161 See-to-touch paper for CoRL
>
> To avoid violating anonymity requirements, we will follow the Program Chair's recommendation to reply to this concern through a different channel.
>
> Thank you again for your detailed review. We believe that the additional discussions and explanations that you have suggested has improved this work. Please let us know if there are any other questions or concerns you may have!
>
> Regards,
>
> Authors of Paper 49.

---

> ### Author Response · Authors · 2023-08-11
> **Regarding More Concerns / Questions**
>
> Thank you once again for dedicating your time to offer detailed feedback on our submission. We would greatly appreciate if you could inform us if there are any further concerns or questions that might hinder a recommendation for acceptance.
>
> Regards,
>
> Authors of Paper49.

---

### Author Response · Authors · 2023-08-08
**General Response to the Reviewers, and List of Updates**

We thank the reviewers for their thoughtful comments and feedback. We are glad that you found our work to be ‘comprehensive’ (Tv7A), ‘well-written and original’ (rRsP), and an ‘important finding to the community’ (27e9). At the same time, several concerns have been raised. In this global response we would like to address a common concern about the use of non-parametric policies along with listing new additions we will be making to paper based on your suggestions.

**Comparing non-parametric approaches with parametric ones** (x3zE, Tv7A, rRsP):

Our framework operates in the low-data regime –  we only have 6 demonstrations - 300 frames in total. Most of the parametric models we use in our baselines require >30 demonstrations for successful policy learning. Owing to the inherent difficulty in teleoperation involving high-dimensional dexterous robotic hands, there exists an insufficient amount of data to effectively train the mentioned offline imitation learning algorithms. Because of this we observe overfitting from these models on the small number of demonstrations.

To ensure fairness in evaluation we have used official implementations from the public Github repositories along with running internal hyper parameter search. We are happy to share our hyperparameters and code if needed. Finally, a more detailed discussion on the limitations of non-parametric approaches will be added to the limitations section.

**Changelog proposed**:

According to the reviews received, we will make 4 changes for the final version of the paper:

1. **Detailed figure for the tactile setup**: We will include a new figure to explain the mapping of the tactile readings from the hand to the images that we use, to the supplementary material and include it for explaining the structure of the tactile pads.
2. **Figure for the contribution of the tactile pads for each task**: We will include the heat map of the variance of each taxel during each task to the supplementary material and mention it in the paper.
3. **Further explanation of a few figures and tables**: We will include more detailed information on Figures 4,5,7 and Table 2.
4. **Expansion of the Experiments section**: Introduction to teleoperation system and downstream tasks will be added to the Experiments section.

We greatly appreciate all the comments and suggestions from the reviewers. Please feel free to reach out to us if you have any additional questions, comments or suggestions!

Regards,

Authors of Paper 49.

---

### Decision · Program_Chairs · 2023-08-30

**Decision:**

Accept (Poster)

**Comment:**

The paper presents a method for pretraining tactile representations using unstructured play data. When combined with visual observations, this method demonstrates improved performance in fine-grained manipulation tasks through imitation learning, using just a handful of demonstrations.

The overall sentiment from the reviewers regarding this paper is predominantly positive. They recognize the significance of the tasks undertaken, the clarity of the method's description, the breadth of real-world evaluations, the importance of the findings in tactile dexterous manipulation, and the anticipated usefulness of the dataset to the community.

However, some reviewers raised concerns about the novelty of the proposed approach, the actual dexterity of the tested tasks, and the scalability of the nearest neighbor method. While the community broadly acknowledges that tactile information enhances complex contact-rich manipulation tasks and play data can facilitate the learning process, this paper primarily offers a systems contribution. Given the capabilities showcased in the evaluations, the detailed system design, and the key findings, this work can be of great value to the community. As a result, I am inclined towards acceptance.

The authors have promised a number of changes to the final version of the paper. Please make sure to include those in your revision.

Additionally, Reviewer x3zE expressed concerns regarding a perceived similarity between this paper and another submission, No. 161 See-to-touch, also under review at CoRL. As the Area Chair for Paper 49, I do not have access to Paper 161 for a detailed comparison. And the discussion of the resemblance might violate anonymity requirements. The Program Chair recommends handling this concern through a different channel, so the final decision on this issue rests with the PC.